# The Individual and Combined Effects of Social Networks and Loneliness on Life Satisfaction among Community-Dwelling Residing Older Adults: A Longitudinal Study

**DOI:** 10.3390/healthcare11070935

**Published:** 2023-03-24

**Authors:** Hui Foh Foong, Rahimah Ibrahim, Tengku Aizan Hamid, Mohamad Fazdillah Bagat

**Affiliations:** 1Malaysian Research Institute on Ageing (MyAgeingTM), Universiti Putra Malaysia, Serdang 43400, Selangor, Malaysia; huifoh@upm.edu.my (H.F.F.); aizan@upm.edu.my (T.A.H.); fadilbagat@upm.edu.my (M.F.B.); 2Department of Human Development and Family Studies, Faculty of Human Ecology, Universiti Putra Malaysia, Serdang 43400, Selangor, Malaysia

**Keywords:** social networks, loneliness, quality of life, longitudinal, older adults

## Abstract

Social networks and loneliness are correlates of life satisfaction in old age. However, the evidence of the combined effects of social isolation and loneliness on life satisfaction is lacking; therefore, this study also aimed to investigate the combined effects of social networks and loneliness on life satisfaction in Malaysian older adults. Data from two waves of the “Neuroprotective Model for Healthy Longevity among Malaysian Older Adults” study were extracted. The first wave of data collection was completed in February 2013, while the second wave was conducted three years after the first wave. The main statistical analysis used was multivariable logistic regression. For individual effect, social networks (B = 0.375, *p* = 0.007), but not loneliness (B = −0.178, *p* = 0.368) significantly determined life satisfaction. Increasing social network size causes increasing life satisfaction. For combined effects, those in “*the lone farmers*” group (B = 0.279, *p* = 0.044) and “*the majority*” group (B = −0.413, *p* = 0.004) were linked to life satisfaction. Social engagement in older people is important for wellbeing in later life. Therefore, community programs and investing in a quality relationship should be encouraged to obtain adequate support and ultimately promote higher life satisfaction.

## 1. Introduction

### 1.1. Life Satisfaction: An Important Wellbeing Concept in Ageing

People’s life expectancy is increasing today due to advances in medical technology and improved lifestyles, resulting in a rise in the proportion of older people globally. The world’s population is aging, with the 65+ age group growing the fastest. In 2050, one in six people in the world will be over 65 years old. The number of people aged 80 and over is also expected to triple, from 143 million in 2019 to 426 million in 2050 [1]. In Malaysia, it is projected that by 2022, the share of the population aged over 65 years will be 7.9%, an increase from 7.4% in the previous year. Malaysia is expected to have an aging population by 2050 when 15% of its total population will be aged above 65 [2]. This rapid growth will have implications on the subjective wellbeing of older people, in general, and for life satisfaction, in particular.

Life satisfaction is subjective well-being and is regarded as the best indicator of the quality of life [3]. Satisfaction with life is a kind of general and deep inner happiness that emanates from individual experiences in the outside world. In other words, it expresses the positive attitude of the individual towards his life and reflects the individual’s feelings about his past, present, or future [4]. Past studies have concluded several important correlates of life satisfaction; some of them were higher education, higher financial security, higher financial satisfaction, good physical health, and good mental health [5,6,7]. Life satisfaction in old age should be given extra attention as research has shown that it contributed to several indicators of psychosocial well-being, health behaviors, and physical health outcomes [8]. As these domains are known to impact health, life satisfaction also is proven to predict mortality and morbidity in older adults [9,10].

### 1.2. Social Isolation and Loneliness: Distinct but Related

The definition of a social network involves a personal and emotional feeling of inclusion, acceptance, affection, importance, admiration, and necessity from another individual [11]. The structure (network size, proximity), function (exchange of support when needed), and quality of social relations (individual experience of the network) contribute significantly to old age [12]. Older people with smaller social networks are at risk of social isolation. Social isolation is the lack of social contact and objective physical separation from other people, whereas loneliness is the distressing feeling of being alone or separated [13]. Besides changes in the family structure such as the death of a spouse and children moving out from home [14], other factors associated with social isolation and loneliness in older people can be due to age-related changes such as sensory loss, cognitive impairment, disability, mobility problems, and multimorbidity [15]. According to a study in Malaysia, by using the Lubben Social Network Scale, almost half (49.8%) of older people living in the community were at risk for social isolation [16]. Although the information on the prevalence of loneliness in Malaysian older adults was limited, a local study has reported that higher loneliness was found to be associated with being Malay, older age, being non-married, lower education level, lower household income, poor health status, and higher physical limitations; however, feelings of loneliness were negatively associated with co-residence with adult children and participation in religious activities [17]. This finding that nearly half of Malaysia’s older adults were at risk of social isolation is concerning, as social isolation and loneliness can lead to other serious medical conditions. For example, a recent study reported that social isolation in later life is associated with an increased risk of depression in England and Japan [18]. Also, as reported in the US, social isolation and loneliness are associated with an 8.0% and 5.0% higher risk for incident cardiovascular disease, respectively, after adjusting for potential covariates [19]. To make things worse, longitudinal studies also confirmed that social isolation [20,21,22] and loneliness [20,21] were associated with all-cause mortality in older people.

### 1.3. The Effects of Social Isolation and Loneliness on Life Satisfaction in Older Adults

Although social isolation and loneliness are different concepts, past literature reported that they were associated with life satisfaction. For example, a multinational study comparing the association between social networks and life satisfaction across different regions of Europe found that the positive relationship between network size and life satisfaction was consistent across countries [23]. This study indicated that a larger social network is important for older adults to be more satisfied with life. Besides, in urban areas of India, a study found that different types of social networks act differently on the perception of life satisfaction among the older person as networks with family, neighbors, friends, and close ones were significantly associated with higher life satisfaction of the older people [24]. This study indicated that support derived from different social networks is important for life satisfaction at later ages.

According to a recent longitudinal study in Hong Kong that involved a community sample of 200 older adults, higher levels of loneliness were found to be associated with lower life satisfaction, and this negative relationship was attributed to reduced social participation [25]. This study further highlighted the importance of social networks and social participation in contributing to the life satisfaction of older adults. In Poland, Szcześniak and colleagues conducted a study to understand how loneliness was linked to life satisfaction, and whether self-esteem and lifelong learning can alter this relationship. The results reported that higher levels of loneliness were associated with lower levels of life satisfaction. Interestingly, this relationship was mediated by self-esteem and moderated by educational activities [26]. This study concluded that although loneliness and social isolation have a negative association with the life satisfaction of older adults, this relationship may be altered by empowering older people’s self-esteem through their involvement in lifelong learning.

### 1.4. Sociodemographic and Economic Correlates of Life Satisfaction in Older Adults

Some demographic variables are important determinants of life satisfaction in old age. For example, according to a recent study, older age, higher education attainment, and higher self-rated health were associated with higher life satisfaction [27]. However, sex, marital status, and employment status did not predict life satisfaction in this study [27]. In addition, another study also demonstrated similar results, reporting that advancing age and higher economic conditions were associated with higher life satisfaction in old age [28]. Contrary to this, the study showed that sex did have a relationship with life satisfaction, as it reported that women had higher life satisfaction than men [28]. In Malaysia, a recent study reported that those in the younger age group (60–74 years old), being male, and residing in an urban area had significantly higher wellbeing [29].

### 1.5. Theory Underpinning the Study

The theory that underpins this study is the Socioemotional Selectivity Theory. This theory posits that as people grow older and their time starts to be perceived as limited, their motivational orientation begins to change [30]. The changes include social network size as it tends to be smaller as the relationship goal has shifted from knowledge-based to emotional-based. In other words, older people’s social networks tend to shrink as they are very selective in socialization and only build relationships with important others [31]. Wellbeing is also affected by such a goal shift as they experience negative or positive emotions if something obstructs their goals or if their goals have been attained [32]. In this study, we examined the changes in social networks and examined whether the changes in social networks in older age could affect their life satisfaction.

### 1.6. The Present Study

Although the concepts of loneliness and social isolation are often discussed and compared, they are largely examined separately, even if they are both included in the same study [33]. Examining the combined effects of social isolation and loneliness on life satisfaction allows us to identify older people who might be at risk of experiencing poor mental health and who may benefit from interventions tailored to their needs in the future. Considering social networks and loneliness together will aid in the understanding of the social situation of older people and provide new directions for research and intervention programs for older people, as well as information relevant to the intersectional effects of social isolation and loneliness on life satisfaction in an Asian nation. This study aims (i) to determine the individual effects of social networks and loneliness on life satisfaction and (ii) to determine the combined effects of social networks and loneliness on life satisfaction among community-dwelling older people by using representative national longitudinal data. Given the general scarcity of studies relating to the intersectional effects of social isolation and loneliness on life satisfaction in an Asian nation, hypotheses were developed based on the fact that social isolation and loneliness were negatively associated with life satisfaction. Hypotheses postulated were, for individual effects, that increased social networks (H_A_1) and decreased loneliness (H_A_2) are associated with increased life satisfaction, respectively. While for the combined effects, the intersection of decreased social networks and increased loneliness is associated with decreased life satisfaction (H_A_3).

## 2. Materials and Methods

### 2.1. Participants and Data Collection

Baseline and second-wave data of a longitudinal study in Malaysia, “Identifying Psychosocial and Identifying Economic Risk Factor of Cognitive Impairment among Elderly” were used. The study included representative community-dwelling older adults in Peninsular Malaysia who were randomly selected through a multistage proportional cluster random sampling technique. The inclusion criteria of this study were adults aged 60 or above, Malaysian nationality, absence of severe mental disorders such as dementia, and gave consent to participate in the study. Those respondents with an MMSE score of 14 or below were excluded because this indicated moderately severe to severe cognitive impairment. The response rate of the first wave of data collection was 87.8%, and the study team managed to follow up on 52.3% of them after three years. The first wave of data collection was conducted from May 2012 until February 2013, while the second wave was conducted three years after the first wave. The data collection was conducted through face-to-face interviews by trained enumerators using a semi-structured questionnaire. The questionnaire consisted of various variables; however, only the variables on demographic, health-related variables, social networks, and loneliness were of special interest in this study. The variables of interest were measured at a baseline and a follow-up after three years.

### 2.2. Measurements

a. Quality of life (dependent variable)—The single-item quality of life measure measured quality of life. Respondents were asked, “Are you satisfied with your life”? with a 4-point scale: 1 (very satisfied), 2 (satisfied), 3 (not satisfied), and 4 (not satisfied at all) [34]. This item was reverse coded before data analysis such that higher values represented higher life satisfaction. The changes in life satisfaction were calculated by T2-T1, and scores were further categorized into two groups for multiple logistic regression analysis; group 1: increased life satisfaction and group 2: no change or decreased life satisfaction. In total, 620 respondents reported no change in their quality of life. A study showed that the single-item life satisfaction measure performed very similarly to the multiple-item satisfaction with life scale (SWLS) and demonstrated a substantial degree of criterion validity with the SWLS [35].

b. Social networks (independent variable)—Social network size was measured via Lubben Social Network Scale-6 [36]. This scale measures the social network size by asking how many relatives and friends they see, feel comfortable talking to about private matters, and feel close to such that they could call on them for help. Higher frequency indicated a larger social network. The changes in social networks were calculated by T2-T1, and scores were further categorized into two groups for multiple logistic regression analysis; group 1: increased social networks and group 2: no change or decreased social networks. A total of 25 respondents reported no change in social networks. The scale has been proven feasible to be used among local community-dwelling older adults in a previous study [37].

c. Loneliness (independent variable)—Loneliness was assessed with a three-item loneliness scale [38]. Each item is rated on a 3-point Likert scale from 1 to 3, asking respondents how often they feel lacking in companionship or being left out and isolated from others. A higher score indicates a higher level of loneliness. Overall, this scale demonstrates high internal consistency for T1 (Cronbach’s α = 0.93) and T2 (Cronbach’s α = 0.96). The changes in social networks were calculated by T2-T1, and scores were further categorized into two groups for multiple logistic regression analysis; group 1: increased loneliness and group 2: no change or decreased loneliness. In total, 923 respondents reported no change in loneliness. The scale has been proven feasible to be used among local community-dwelling older adults in a previous study [39].

d. Background characteristics (control variables)—Background variables included gender, age, education level, marital status, ethnicity, poverty status, objective health status, and subjective health status. Non-married consisted of those widowed, divorced, separated, and never married. Non-Malays were those of non-native, Chinese, and Indian ethnicities. Aside from that, the poverty status was determined by Malaysia’s poverty line income. Respondents with a household income of less than MYR2,629 were considered living in poverty or in the bottom 40% of Malaysians’ household income, also known as the B40 group [40,41].

### 2.3. Categorization of Social Networks and Loneliness

Participants were categorized into four groups based on their changes in social networks and loneliness [33]. They were: group 1: “*the vulnerable*”—no change or decreased social networks intersected with increased loneliness; group 2: “*the lone farmers*”—no change or decreased social networks intersected with no change or decreased loneliness; group 3: “*lonely in a crowd*”—increased social network intersected with increased loneliness; and group 4: “*the majority*”—increased social networks intersected with no change/decreased loneliness [33].

### 2.4. Data Preparation and Analytic Strategies

Attrition is unavoidable in longitudinal studies. In this study, the rate of those unavailable to follow up for loneliness, social networks, and quality of life during T2 was 45.8% (*n* = 1037), 45.6% (*n* = 1023), and 45.5% (*n* = 1028), respectively. Although the attrition rate was less than 50%, missing values should be replaced as they could cause selection and data bias. In the analysis, multiple imputations were used to replace missing values [42]. Social networks, loneliness, and life satisfaction in T2 were imputed for participants with missing data. The impact of events in the two-wave data could be evaluated through the lagged dependent variable or change score methods; Johnson concluded that the change score method has advantages over the lagged dependent variable technique [43]. The change scores for life satisfaction, social networks, and loneliness were calculated by subtracting T1 scores from T2 scores. Higher change scores indicate an increase in life satisfaction, social networks, and loneliness. First, the chi-square statistic was analyzed to identify the association between the main study variables and the combination of main study variables with gender and quality of life. Then, multivariate logistic regression analysis was analyzed to identify the individual and the combined effects of social networks and loneliness on quality of life. All the logistic regression analyses involved unadjusted (control variables were not considered) and adjusted models (models involved control variables—gender, age, ethnicity, education level, marital status, poverty status, changes in chronic condition, and changes in self-rated health status). The significance level was set at 5%, and IBM SPSS Statistics Version 24 was used to perform all the statistical analyses.

## 3. Results

### 3.1. Background Characteristics of the Sample

Among the 2315 respondents for T1, 1211 were included in the T2 survey, indicating an inclusion rate of 52.3%. Table 1 presents the background characteristics of the sample. From the 1211 respondents in T2, 600 (49.5%) were male and 611 (50.5%) were female. Additionally, most of them were from a younger age group of 60–69 years old (*n* = 748, 61.8%), while the rest of them were of an older age group (aged 70 and above) (*n* = 463, 38.2%). Most of the respondents were native Malay (*n* = 768, 63.6%), and 440 (36.4%) were non-native (Chinese and Indian). In terms of marital status, most of them were married (*n* = 864, 71.3%), and 347 (28.7%) were non-married; either single, divorced, separated, widow, or widower. Most respondents received at least primary education (*n* = 981, 81.0%), and 230 (19.0%) of them did not receive formal education. In terms of poverty status, 57.6% (*n* = 685) of them lived in poverty during the baseline data collection. For health status, 449 (37.4%) of them reported that they had increased chronic medical conditions. However, 942 (78.7%) of them said to have no change/decrease in perceived health status. In total, 39.2% (*n* = 469) of them reported having increased social networks, 15.9% (*n* = 191) reported having increased loneliness, and lastly, 26.4% (*n* = 320) of them reported having increased quality of life.

### 3.2. Distribution of Main Study Variables by Gender and Quality of Life

The chi-square statistic was analyzed to identify the association between changes in main study variables with gender and quality of life. As depicted in Table 2, there was a significant association between social networks and gender (χ^2^ = 4.317, *p* = 0.038). More women (*n* = 254, 42.1%) than men (*n* = 215, 36.3%) reported having increased social networks. However, there was no significant association between loneliness (χ^2^ = 0.182, *p* = 0.669) and quality of life (χ^2^ = 0.110, *p* = 0.740) with gender. In addition, there was also no significant association found between the pattern of social networks and loneliness with gender (χ^2^ = 4.483, *p* = 0.214).

Next, the chi-square statistic identified a significant association between social networks and quality of life (χ^2^ = 9.693, *p* = 0.002). The majority of those who reported having no change/decrease in quality of life reported having no change/decrease in social networks (*n* = 556, 63.5%), more than those who reported an increase in quality of life (*n* = 167, 53.5%). In addition, a significant association was also found between the pattern of social networks and loneliness with quality of life (χ^2^ = 11.284, *p* = 0.010). In total, 52.1% (*n* = 451) of those who reported no change/decrease in social network and loneliness reported no change/decrease in quality of life compared to those who had increased quality of life (*n* = 137, 44.5%). In addition, 41.9% (*n* = 129) of those who reported increased quality of life reported having increased social networks and no change/decrease in loneliness, more than those with no change/decrease in quality of life (*n* = 271, 31.3%). However, no significant association was noted between loneliness and quality of life (χ^2^ = 1.650, *p* = 0.199). Moreover, the effect size of the association between loneliness and quality of life was negligible (φ = −0.037).

### 3.3. Individual Effects of Social Networks and Loneliness on Quality of Life

Multiple logistic regression was analyzed to identify the individual effects of social networks and loneliness on quality of life. As reported in Table 3, in the unadjusted model, when control variables were not included, only social networks had a significant relationship with quality of life (B = 0.387, *p* = 0.004, odd ratio = 1.473). After controlling for potential covariates, social networks still appeared to have significant effects on quality of life (B = 0.375, *p* = 0.007, odd ratio = 1.455). Those who had no change/decrease in social networks had approximately 1.5 times higher risk of experiencing no change/decrease in quality of life than those who reported an increase in social networks. However, no significant relationship was identified between loneliness and quality of life in both unadjusted (B = −0.200, *p* = 0.292, odd ratio = 0.818) and adjusted (B = −0.178, *p* = 0.368, odd ratio = 0.837) models. In terms of covariates, only ethnicity (B = −0.634, *p* < 0.001, odd ratio = 0.531) and changes in perceived health status (B = 0.377, *p* = 0.020, odd ratio = 1.458) were significantly associated with the quality of life. Being Malay was associated with a 47% lower risk of reporting no change/decrease in quality of life. Those who reported having no change/decrease in perceived health status had approximately 1.5 times higher risk of no change/decrease in quality of life.

### 3.4. Combined Effects of Social Networks and Loneliness on Quality of Life

Multiple logistic regression was also analyzed to identify the combined effects of social networks and loneliness on quality of life. Four different combinations of social networks and loneliness (the four groups) were included in four multiple logistic regression models to examine which pattern of social networks and loneliness affect the quality of life. As reported in Table 4, in both unadjusted (B = 0.307, *p* = 0.021, odd ratio = 1.360) and adjusted (B = 0.279, *p* = 0.044, odd ratio = 1.321) models, “*the lone farmers*” group was associated with quality of life. After controlling for covariates, “*the lone farmers*” group had a 1.3 times higher risk of developing no change/decreased quality of life. In addition, “*the majority*” group was also associated with changes in quality of life in both adjusted (B = −0.457, *p* = 0.001, odd ratio = 0.633) and unadjusted models (B = −0.413, *p* = 0.004, odd ratio = 0.662). After controlling for covariates, “*the majority*” group had a 34% lower risk of developing no change/decrease in quality of life.

## 4. Discussion

### 4.1. The Individual and Combined Effects of Social Networks and Loneliness on Life Satisfaction

The H_A_1 of this study is supported as the present study found that increased social networks were associated with an elevation in the quality of life in both the unadjusted and adjusted models. On top of that, those in “*the lone farmers*” group were also found to have a higher tendency to report no change or decrease in life satisfaction. This finding indicates that social networks are a stronger determinant of life satisfaction than loneliness although, with no change or decreasing loneliness, life satisfaction was still not changed or increased due to the no change or decrease in social networks. The findings indicate that more extensive social networks could promote quality of life in old age. With advancing age, older adults’ social networks tend to shrink. The smaller size of social networks could be due to the death of a spouse, health problems, disability, children leaving home, and more selectivity in choosing social partners to favor emotionally significant relationships over more peripheral ones [44]. Social networks improve the quality of life in older people because they act as an essential resource during crises. A more extensive social network also means more confidants or close ties that allow them to deal with stressful environments or difficult life experiences. The social and emotional support provided by relatives and friends is valuable for older adults during a crisis [45]. For example, according to a study in Ghana, community-dwelling older adults experiencing spousal loss were less likely to be psychologically distressed if they engaged in meaningful social support [46]. In addition, poor social networks in older people are also associated with various negative outcomes such as higher depression symptoms, reduced grip strength, disabilities of intellectual activity and social role, a higher tendency of homebound status, and disability in activities of daily living [47].

However, hypothesis H_A_2 of this study is not supported as this study failed to establish the relationship between loneliness and quality of life in both the unadjusted and adjusted models. The findings are not in line with most previous studies that listed the negative relationship between loneliness and wellbeing [48,49]. The results should be interpreted with caution as it does not mean loneliness is not an essential determinant of life satisfaction. It is just that compared to social networks, which are highly associated with the presence of a confidant and source of aid, loneliness had less weightage on life satisfaction than social networks. This argument is supported by the finding that in Model 2, “*the majority*” group was still linked to an increase in life satisfaction (H_A_3 is supported). The findings could be attributed to the acceptance of Malaysian older people of loneliness in old age. They acknowledge that loneliness is always associated with old age due to the death of a spouse, children moving away from home, deterioration in health, disability, and mobility problems. Therefore, they can adapt to the changes well without affecting their life satisfaction. This argument is supported by Selective Optimization with Compensation Theory where older people always adapt to psychological changes related to changes in family structure and health conditions due to aging [50].

### 4.2. Socio-Demographic and Economic Determinants of Life Satisfaction

This study also found that ethnicity was significantly associated with life satisfaction. Specifically, Malay individuals were more likely to report increased life satisfaction compared to non-Malay individuals. The relationship between ethnicity and life satisfaction could be attributed to Malay older people having a higher level of wellbeing than non-Malay older adults in Malaysia. This argument is supported by a local study that found that non-Malay older adults in Malaysia had a higher depression risk compared to Malay older people [51].

In addition, self-rated health status was also significantly associated with life satisfaction. Specifically, those who reported having no change or decreased self-rated health status were at a higher tendency of no change or reduced life satisfaction. The findings are similar to a study by Dumittrache and colleagues in which they discovered that perceived health status was positively associated with life satisfaction in Spain’s older adults [52]. Excellent self-rated health status is often associated with the absence of disability and pain; therefore, it contributes to a higher level of wellbeing. However, good self-rated health status is not all about good things in older adults. Good self-rated health often stops older people from getting medical check-ups as they perceived their health condition as good; ultimately it can lead to a high possibility of underdiagnoses of some major illnesses. People tend not to seek medical check-ups or treatment until they feel certain pain or symptoms that could affect their daily living. This argument is supported by a study in Singapore that found that the utilization of inpatient and outpatient medical facilities is high even among older people that reported high self-rated health status [53].

### 4.3. Limitations and Strengths of the Study

There are several limitations of the study that need to be acknowledged: (i) although we used a widely used and well-developed scale for measuring loneliness, the measurement of loneliness remains a challenge. Loneliness is a negative construct, causing social desirability bias as respondents had difficulty reporting feeling lonely. Thus, loneliness might be underreported in this study; (ii) the data collection only involved Peninsula Malaysia, disturbing the generalization of findings to other settings such as East Malaysia and aged care institutions. It would be interesting to reproduce the study in East Malaysia and aged care institutions for future research to compare and further validate the findings; (iii) this study also did not include the measurement of the quality of relationships. Past studies reported that instead of only a quantitative measure of social networks, the qualitative measure of the relationship (e.g., the quality of the relationship) is also important in determining life satisfaction [54]. Therefore, future studies should also consider including a qualitative measure of social networks (e.g., relationship satisfaction) to compare the effects of quantitative and qualitative measures of social networks on wellbeing in older people. Moreover, we classified respondents a priori into four groups. Therefore, this causes the sample size of each group to be very different, and this might affect the interpretation of the results. It would be useful in future studies to determine if the same groups emerge through cluster analysis. Furthermore, major life events such as the death of a spouse and retirement have an impact on wellbeing. Therefore, future studies should consider evaluating the combined effects of social isolation and loneliness before and after such major life events to determine if they have a different effect on wellbeing.

This study had several strengths; first, this study utilized a longitudinal research design, meaning that the relationship between independent and dependent variables could be interpreted as causative. Second, all the analyses consider controlling a series of control variables; therefore, the relationships between variables could be estimated more accurately with minimal bias.

## 5. Conclusions

This study reported three significant findings; (i) the reduction in social networks was associated with a decrease in life satisfaction; (ii) respondents who were in “*the lone farmers*” group had a higher tendency of no change or a decrease in life satisfaction; and (iii) respondents who were in “*the majority group*” had a higher tendency of increased life satisfaction. The study provides theoretical implications where it adds to the literature on Socioemotional Selectivity Theory that older adults might not require extensive social networks to maintain their wellbeing; instead, a high-quality relationship that could eliminate the feeling of loneliness is more important in promoting wellbeing. In terms of practical implications, the findings suggest that social engagement in older people is important for wellbeing in later life. Older persons should continue to participate in age-appropriate activities because social interaction can help them stay motivated, realize their full potential, and ultimately find more life satisfaction. Therefore, community programs such as special aids, and befriending programs for older people are needed to promote life satisfaction. Moreover, investing in a quality relationship should be started and highly encouraged at a younger age to obtain adequate support at an older age and ultimately promote high life satisfaction.

## Figures and Tables

**Table 1 healthcare-11-00935-t001:** Background characteristics of the sample (*n* = 1211).

		*n*	%
Gender	Male	600	49.5
	Female	611	50.5
Age	Younger (60–69 years old)	748	61.8
	Older (70 years old and older)	463	38.2
Ethnicity	Malay	768	63.6
	Non-Malay	440	36.4
Education level	Formal education	981	81.0
	No formal education	230	19.0
Marital status at T1	Married	864	71.3
	Non-married	347	28.7
Poverty status at T1	B40 group	685	57.6
	Non B40 group	505	42.4
Changes in number of chronic disease(s)	No change	753	62.6
	Increase	449	37.4
Changes in perceived health status	No change/decrease	942	78.7
	Increase	255	21.3
Social networks	No change/decrease	727	60.8
	Increase	469	39.2
Loneliness	No change/decrease	1013	84.1
	Increase	191	15.9
Life satisfaction	No change/decrease	891	73.6
	Increase	320	26.4

Note: T1, baseline; *n*, frequency; %, percentage; B40, bottom 40% of the Malaysian household income.

**Table 2 healthcare-11-00935-t002:** Distribution of main study variables by gender and life satisfaction.

		Gender		Chi-Square Value	Effect Size	*p*-Value
		Male, *n* (%)	Female, *n* (%)			
Social network	Increase	215 (36.3)	254 (42.1)	4.317	−0.060 ^a^	**0.038**
	No change/decrease	378 (63.7)	349 (57.9)			
Loneliness	Increase	505 (84.6)	508 (83.7)	0.182	−0.012 ^a^	0.669
	No change/decrease	92 (15.4)	99 (16.3)			
Life satisfaction	Increase	156 (26.0)	164 (26.8)	0.110	−0.010 ^a^	0.740
	No change/decrease	444 (74.0)	447 (73.2)			
−social network and −loneliness (“*The lone farmers*”)		309 (52.7)	280 (47.1)	4.483	0.062 ^b^	0.214
+ social network and −loneliness (“*The majority*”)		188 (32.1)	217 (36.5)			
−social network and +loneliness (“*The vulnerable*”)		62 (10.6)	62 (10.4)			
+ social network and +loneliness (“*Lonely in a crowd*”)		27 (4.6)	35 (5.9)			
		Life satisfaction		Chi-square value	Effect size	*p*-value
		Increase, *n* (%)	No change/decrease, *n* (%)			
Social network	Increase	145 (46.5)	319 (36.5)	9.693	0.090 ^a^	**0.002**
	No change/decrease	167 (53.5)	556 (63.5)			
Loneliness	Increase	43 (13.6)	147 (16.7)	1.650	−0.037 ^a^	0.199
	No change/decrease	273 (86.4)	734 (83.3)			
−social network and −loneliness (“*The lone farmers*”)		137 (44.5)	451 (52.1)	11.284	0.098 ^b^	**0.010**
+ social network and −loneliness (“*The majority*”)		129 (41.9)	271 (31.3)			
−social network and +loneliness (“*The vulnerable*”)		28 (9.1)	95 (11.0)			
+social network and +loneliness (“*Lonely in a crowd*”)		14 (4.5)	48 (5.5)			

Note: −, no change/decrease; +, increase; *n*, frequency; bold value, *p* < 0.05; ^a^, Phi (φ) as the effect size in 2 × 2 contingency table; ^b^, Cramer’s V as the effect size in bigger table.

**Table 3 healthcare-11-00935-t003:** Individual effects of social networks and loneliness on life satisfaction.

DV—Life Satisfaction (0—Increase, 1—No Change/Decrease)	B	S.E.	*p*-Value	Exp (B)	95% C.I for Exp (B)	
					Lower	Upper
**Unadjusted model ^a^**						
Social network (0—increase, 1—no change/decrease)	0.387	0.134	0.004	1.473	1.132	1.917
Loneliness (0—increase, 1—no change/decrease)	−0.200	0.190	0.292	0.818	0.563	1.189
**Adjusted model ^b^**						
Gender (0—male, 1—female)	−0.013	0.159	0.932	0.987	0.723	1.347
Age (0—60–69 years old, 1—70 and above)	−0.071	0.149	0.634	0.931	0.696	1.247
Ethnicity (0—non-Malay, 1—Malay)	−0.634	0.155	**<0.001**	0.531	0.392	0.718
Education level (0—attended formal education, 1—no formal education)	−0.088	0.193	0.648	0.916	0.628	1.336
Marital status (0—married, 1—non-married)	0.091	0.175	0.602	1.096	0.778	1.543
Poverty status (0—non-B40, 1—B40)	−0.033	0.145	0.821	0.968	0.728	1.287
Number(s) of chronic condition (0—increase, 1—no change/decrease)	0.123	0.142	0.387	1.130	0.856	1.492
Perceived health status (0—increase, 1—no change/decrease)	0.377	0.162	**0.020**	1.458	1.061	2.003
Social network (0—increase, 1—no change/decrease)	0.375	0.139	**0.007**	1.455	1.107	1.912
Loneliness (0—increase, 1—no change/decrease)	−0.178	0.198	0.368	0.837	0.567	1.234

Note: ^a^, model did not include demographic variables as control variables; ^b^, model included demographic variables as control variables; DV, dependent variable; B, unstandardized coefficient; S.E, standard error; Exp (B), odd ratio; C.I, confidence interval; B40, bottom 40% of the Malaysian household income; 0, reference group; 1, non-reference group; bold value, *p* < 0.05.

**Table 4 healthcare-11-00935-t004:** Combined effects of social network and loneliness on life satisfaction.

**Model 1: −social network and −loneliness on life satisfaction (“*The lone farmers*”)**						
DV—Life satisfaction (0—increase, 1—no change/decrease)	B	S.E.	*p*-value	Exp (B)	95% C.I for Exp (B)	
					Lower	Upper
**Unadjusted model ^a^**						
−social network and −loneliness (0—no, 1—yes)	0.307	0.133	**0.021**	1.360	1.047	1.766
**Adjusted model ^b^**						
Gender (0—male, 1—female)	−0.026	0.158	0.870	0.975	0.715	1.329
Age (0—60–69 years old, 1—70 and above)	−0.067	0.149	0.654	0.936	0.699	1.252
Ethnicity (0—non-Malay, 1—Malay)	−0.637	0.154	**<0.001**	0.529	0.391	0.715
Education level (0—attended formal education, 1—no formal education)	−0.062	0.192	0.745	0.939	0.644	1.369
Marital status (0—married, 1—non-married)	0.094	0.174	0.591	1.098	0.780	1.546
Poverty status (0—non-B40, 1—B40)	−0.018	0.145	0.900	0.982	0.739	1.305
Number(s) of chronic condition (0—increase, 1—no change/decrease)	0.119	0.141	0.400	1.127	0.854	1.486
Perceived health status (0—increase, 1—no change/decrease)	0.365	0.162	**0.024**	1.441	1.049	1.979
−social network and −loneliness (0—no, 1—yes)	0.279	0.138	**0.044**	1.321	1.008	1.733
**Model 2: +social network and −loneliness on life satisfaction (“*The majority*”)**						
DV—Life satisfaction (0—increase, 1—no change/decrease)	B	S.E.	*p*-value	Exp (B)	95% C.I for Exp (B)	
					Lower	Upper
**Unadjusted model ^a^**						
+social network and −loneliness (0—no, 1—yes)	−0.457	0.137	**0.001**	0.633	0.484	0.828
**Adjusted model ^b^**						
Gender (0—male, 1—female)	−0.015	0.159	0.925	0.985	0.722	1.344
Age (0—60–69 years old, 1—70 and above)	−0.070	0.149	0.640	0.933	0.696	1.249
Ethnicity (0—non-Malay, 1—Malay)	−0.630	0.154	**<0.001**	0.533	0.394	0.721
Education level (0—attended formal education, 1—no formal education)	−0.078	0.193	0.685	0.925	0.634	1.349
Marital status (0—married, 1—non-married)	0.090	0.175	0.604	1.095	0.777	1.541
Poverty status (0—non-B40, 1—B40)	−0.030	0.145	0.834	0.970	0.730	1.289
Number(s) of chronic condition (0—increase, 1—no change/decrease)	0.120	0.142	0.396	1.128	0.854	1.489
Perceived health status (0—increase, 1—no change/decrease)	0.372	0.162	**0.022**	1.450	1.056	1.992
+social network and −loneliness (0—no, 1—yes)	−0.413	0.141	**0.004**	0.662	0.501	0.873
**Model 3: −social network and +loneliness on life satisfaction (“*The vulnerable*”)**						
DV—Life satisfaction (0—increase, 1—no change/decrease)	B	S.E.	*p*-value	Exp (B)	95% C.I for Exp (B)	
					Lower	Upper
**Unadjusted model ^a^**						
−social network and +loneliness (0—no, 1—yes)	0.210	0.226	0.353	1.234	0.792	1.922
**Adjusted model ^b^**						
Gender (0—male, 1—female)	−0.029	0.158	0.856	0.972	0.713	1.325
Age (0—60–69 years old, 1—70 and above)	−0.065	0.149	0.664	0.937	0.700	1.254
Ethnicity (0—non-Malay, 1—Malay)	−0.648	0.154	**<0.001**	0.523	0.387	0.707
Education level (0—attended formal education, 1—no formal education)	−0.079	0.192	0.682	0.924	0.634	1.348
Marital status (0—married, 1—non-married)	0.074	0.174	0.672	1.077	0.765	1.514
Poverty status (0—non-B40, 1—B40)	−0.049	0145	0.735	0.952	0.717	1.264
Number(s) of chronic condition (0—increase, 1—no change/decrease)	0.125	0.141	0.377	1.133	0.859	1.494
Perceived health status (0—increase, 1—no change/decrease)	0.384	0.162	**0.018**	1.468	1.069	2.015
−social network and + loneliness (0—no, 1—yes)	0.268	0.237	0.258	1.308	0.821	2.082
**Model 4: +social network and +loneliness on life satisfaction (“*Lonely in a crowd*”)**						
DV—Life satisfaction (0—increase, 1—no change/decrease)	B	S.E.	*p*-value	Exp (B)	95% C.I for Exp (B)	
					Lower	Upper
**Unadjusted model ^a^**						
+social network and +loneliness (0—no, 1—yes)	0.210	0.311	0.500	1.234	0.670	2.271
**Adjusted model ^b^**						
Gender (0—male, 1—female)	−0.033	0.158	0.833	0.967	0.710	1.318
Age (0—60–69 years old, 1—70 and above)	−0.063	0.149	0.673	0.939	0.702	1.257
Ethnicity (0—non-Malay, 1—Malay)	−0.646	0.154	**<0.001**	0.524	0.288	0.708
Education level (0—attended formal education, 1—no formal education)	−0.064	0.192	0.737	0.938	0.643	1.366
Marital status (0—married, 1—non-married)	0.075	0.174	0.666	1.078	0.767	1.516
Poverty status (0—non-B40, 1—B40)	−0.041	0.144	0.775	0.960	0.723	1.274
Number(s) of chronic condition (0—increase, 1—no change/decrease)	0.122	0.141	0.387	1.130	0.857	1.490
Perceived health status (0—increase, 1—no change/decrease)	0.376	0.161	**0.020**	1.457	1.062	1.999
+social network and +loneliness (0—no, 1—yes)	0.063	0.319	0.843	1.065	0.570	1.991

Note: ^a^, model did not include demographic variables as control variables; ^b^, model included demo-graphic variables as control variables; −, no change/decrease; +, increase; DV, dependent variable; B40, bottom 40% of the Malaysian household income; B, unstandardized coefficient; S.E, standard error; Exp (B), odd ratio; C.I, confidence interval; 0, reference group, 1, non-reference group; bold value, *p* < 0.05.

## Data Availability

The datasets used and/or analyzed during the current study are available from the corresponding author upon reasonable request.

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
