# Peer review of "The Individual and Combined Effects of Social Networks and Loneliness on Life Satisfaction among Community-Dwelling Residing Older Adults: A Longitudinal Study"

_healthcare, 2023, doi:10.3390/healthcare11070935_

Round 1

Reviewer 1 Report

After carefully reading the manuscript entitled ' The Individual and Combined Effects of Social Networks and Loneliness on Life Satisfaction Among Community-Dwelling Residing Older Adults: A Longitudinal Study' I find the authors an acceptable work. However, I have some remarks.

Overall, the manuscript is well written and understandable. Methodologically, the methodology is reproducible even if the authors did not control for desirability bias during the measurement of the variable (loneliness) which is a serious limitation of the study. The presentation of the results can be improved by adding the effect size in the Chi-square analysis. In addition, the formatting of the tables can be improved. Finally, some of the arguments raised in the discussion do not agree with me at all.

Line 140 : ‘The inclusion criteria of this study were adults aged 60 or 140 above, Malaysian nationality, absence of severe mental disorders such as dementia’  How the authors determined this criterion in the participants. I think the authors should clarify whether they used the MMSE or the MoCA or whether they just asked the participants. I think this is very important.

Lines 203-204: ‘Social networks, loneliness, and life satisfaction in T2 were imputed for participants with missing data.’ Can the authors specify the amount of suppressed data?

Line 350-353: ‘The better wellbeing among Malay older people could be attributed to their 350 Islamic spiritual practice as older Muslim practice “redha”, which means to accept the de- 351 cree of the Almighty wholeheartedly. As a result, they are generally content with their 352 lives despite the problems they face while ageing.’

On what basis, the authors believe that religious practice would underlie participants' well-being. At no point in the manuscript did the authors specify the religious practice of the participants. Furthermore, what evidence is there that the same results would not be found among Catholic practitioners or simply among those who have a strong meditation practice? If the authors wish to use this argument, they must absolutely reference their statements with studies that have proven this. In addition, they must provide evidence that all of their participants practice "rheda" and have no other religious practices. 

Reviewer 2 Report

I really enjoyed reading this manuscript. The world's population is ageing, with the 65+ age group growing the fastest. In 2050, one in six people in the world will be over 65 years old. The number of people aged 80 and over is also expected to triple, from 143 million in 2019 to 426 million in 2050. Similarly, the number of centenarians (almost half a million in 2015) will increase markedly during the 21st century, reaching more than 25 million in 2100. Could the authors include in section 1.1. any data on the elderly population in Malaysia?

The difference between satisfactory longevity and unsatisfactory longevity often rests on very simple underpinnings: one of them, perhaps the most important, is the assessment that each person is capable of making of what have been the most important milestones in his or her life.

In this sense, the study is important and relevant since aging in a healthy, active and satisfactory way seems to have become an attainable goal...

The findings suggest that social engagement in older people is important for wellbeing in later life. There are benefits (in functional and cognitive capacity, in self-esteem...) associated with the practice of certain activities as long as the elderly can face new challenges appropriate to their real possibilities, with guarantees, and with the motivation generated by an exciting environment, where they can continue to be motivated by projects, by ideas for renewal, and by the legitimate aspiration for a better life under the essential vitality of illusions. This idea should be included in the conclusions when exemplifying befriendship programmes for older people to promote life satisfaction.

Reviewer 3 Report

Thank you very much for allowing me to read this interesting manuscript. I appreciate the interest of researchers in trying to investigate the Individual and Combined Effects of Social Networks and Loneliness on Life Satisfaction Among Community-Dwelling Residing Older Adults. Nevertheless, there are some comments and recommendations that I would like to make. Improvements are required to the current iteration before publication can be recommended. Please see the below comments and recommendations.

INTRODUCTION

•The introduction is very specific, but does not address all issues equally. It is desirable to go more deeply into the concept of the social network.

•The final paragraph of section 1.4 does not proceed in this section, it is to be expected in the methodology where it is repeated.

•The presentation of the groups is confusing.

•Section 1.5 Three hypotheses are presented, the formulation is not clear: which one is hypothesis 1 and which one is hypothesis 2?

•In order to facilitate reading, it is convenient to end the introduction with the aim of the study

METHODS

•It is not necessary to include information on the coding of the variables. This information is redundant, it only makes it more difficult to read.

•As mentioned above, section 2.3 is repeated with the introduction.

RESULTS

•The presentation of test results is very confusing.

•The tables are very confusing and do not fit standard publication norms.

•Different items have a number of responses in the sample that differs between questions. Due to variability, presenting the total sample size is appropriate.

•It is not necessary to present the codification of the variables, it does not provide information.

•Check table footnotes ¿P<0.005?

•The names of the groups referred to in the manuscript do not correspond to the tables, for better comprehension and interpretation it is necessary to unify the nomenclature.

DISCUSSION

•You have made a subdivision into 4 groups, whereby the sample size of each group is very different, have you assessed whether there are significant differences between the groups? How does this affect the interpretation and discussion of the results obtained?

Reviewer 4 Report

I'm grateful for the opportunity to edit the manuscript titled "The Individual and Combined Effects of Social Networks and Loneliness on Life Satisfaction Among Community-Dwelling Residing Older Adults: A Longitudinal Study " that was submitted to Healthcare, Special Issue - The Effects of Social Relationships on Health in Old Age.
This article needs some changes.
1. The relevance of the research realized by the authors of the paper is obvious but a better rationale is required for publishing a manuscript , besides the results of the study should be of relevance to stakeholders such as researchers, psychologists and social workers, among others. This is lacking in the current paper.
2. Authors mentioned the role of social networks and loneliness in explaining the life satisfaction in old age, however, it would be appropriate to also mention the role of socio-economic and ethno-confessional characteristics.
3. In addition, suggestions for future research should be expanded.
Nevertheless this paper holds actual value to the readers on Healthcare.
I will be glad to review the revised manuscript.

Reviewer 5 Report

Dear editors and authors, first of all, thank you for the opportunity to review this text.

Although your manuscript falls within the aim and scope of this journal, it is being declined due to lack of sufficient novelty.

Author Response

Thank you for the comment; we humbly accept it. 

Reviewer 6 Report

I think this is a very interesting study and well presented. However, I have a few comments that may improve the manuscript. First, the authors lumped together the decrease and no change responses. Although I don't think there is anything inherently wrong with this approach, I would like to know the frequency of responses in the 3 categories: decrease, no change and increase. It may be that the results would have been different if the increase and decrease categories were analysed separately and the no change category removed. Knowing the frequencies would allow one to decide whether lumping together made sense as small numbers in the decrease category could require one to combine categories. 

Another concern I have is in the discussion and conclusions where the authors assume that the use of a longitudinal design allows them to infer causation. This is not true as there may be unmeasured third variables that are responsible for the observed relationships. If the authors are making a claim for causation then they should expand the discussion and explain how this is supported by the evidence. 

Finally, the manuscript needs to be proofread by a native speaker of English. For example, the first sentence in the Introduction the word 'cause' is inserted inappropriately. There are numerous instances throughout the manuscript where the level of English that is used is not acceptable. 

Round 2

Reviewer 1 Report

The authors have responded to my comments. So I give my agreement for the acceptance of their manuscript.

Author Response

Thank you. 

Reviewer 3 Report

The authors have satisfactorily addressed the reviewers's comments however, it still needs a few more revisions (The quantity and clarity of the tables should be improved to make the results understandable and appropriate).

Author Response

We thank the reviewer for their comment on the presentation of the tables. After careful consideration with the team, we feel that the confusion might be due to the adjusted and unadjusted models in Tables 3 and 4. Therefore, we have included descriptions of adjusted and unadjusted models in Tables 3 and 4. Please refer to the parts highlighted in yellow for the amendment. Besides, we have four tables in this manuscript, and the quantity is feasible as allowed by the journal format. Thank you.

Reviewer 5 Report

No additional comments beyond those made in the first review.

Author Response

Thank you. 

Reviewer 6 Report

Although the authors have addressed my concerns, the standard of English is still not acceptable and needs further work, although it has improved. 

Author Response

We thank you so much for the comment. We have carefully examined the English of this manuscript and made some improvements to some sentences. Please refer to the corrections highlighted in green for the amendments. Additionally, we have also submitted this manuscript for a copy-editing service to further enhance the English. Please refer to the attachment for the copy-editing certificate.
